## Article
# Noble Liquid Calorimetry for FCC-ee

**Nicolas Morange** on behalf of the Noble Liquid Calorimeters Study Group

IJCLab, CNRS/IN2P3, Université Paris-Saclay, 91405 Orsay, France; nicolas.morange@cern.ch;
Tel.: +33-1-64-46-83-24

**Abstract:** Noble liquid calorimeters have been successfully used in particle physics experiments for decades. The project presented in this article is that of a new noble liquid calorimeter concept, where a novel design allows us to fulfil the stringent requirements on calorimetry of the physics programme of the electron-positron Future Circular Collider at CERN. High granularity is achieved through the design of specific readout electrodes and high-density cryostat feedthroughs. Excellent performance can be reached through new very light cryostat design and low electronics noise. Preliminary promising performance is achieved in simulations, and ideas for further R&D opportunities are discussed.

**Keywords:** electromagnetic calorimeter; FCC; liquified noble gas

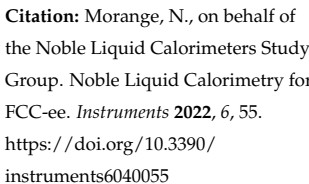

## 1. Introduction

Noble liquid calorimeters have been successfully used in particle physics experiments at colliders for decades, owing to the very good properties of this technology and reasonable construction prices. The latest example is the liquid argon (LAr) sampling calorimeter of the ATLAS experiment at the LHC [1], which has a central role in the whole ATLAS physics programme. The LAr technology allows us to achieve a very good stochastic term of 10% for the resolution of electromagnetic objects and a linearity at the per-mille level over four orders of magnitude, and it has been shown to have gained stability at the $10^{-4}$ level over years of operation.

Following the outcome of the latest European Strategy for Particle Physics that the next high-priority collider should be a Higgs boson factory, CERN is bringing forward the electron-positron Future Circular Collider (FCC-ee) project [2] with its 91 km long ring. The very broad physics programme of the FCC [3], encompassing much more than Higgs boson physics, has strong implications on the design of calorimeters [4]. Indeed, the specifications include excellent hadronic energy resolution (around 4% at 50 GeV for Higgs physics) that in turn points towards highly granular calorimeters optimised for particle flow reconstruction algorithms, very good energy resolution on low-energy photons (for $b$ and $\tau$ physics), excellent shower shape discrimination for $e/\gamma$ against hadrons (in particular for $\tau$ physics), and systematic uncertainties which should be reduced to very low levels. Noble liquid calorimeters have the potential to fulfil all these stringent requirements. As the FCC-ee will feature between two and four interaction points, different detector technologies can be used for future experiments, and noble liquid calorimetry can therefore be the basis of an FCC-ee detector concept.

However, unlike the mature highly granular (CALICE) and dual-readout (DREAM) concepts which have been heavily developed for a decade, the R&D on noble liquid calorimeters stopped twenty-five years ago, and a vigorous effort is therefore needed to prove the feasibility of such a concept. Initial studies towards a high-granular LAr calorimeter were started in the context of the FCC-hh project [5], where noble liquids were seen as the only viable candidates for the electromagnetic calorimeters because of the high radiation levels [6]. The much simpler environment of the FCC-ee (negligible radiation

levels and low data rates) allows us to build upon this design and optimise it towards the ultimate performance required.

## 2. R&D Towards a Granular Noble Liquid Calorimeter at FCC-ee

Achieving the concept of a viable noble liquid calorimeter for FCC-ee requires several specific detector developments, which are in progress. These go in parallel with simulation studies that guide the design and provide the expected performance of the concept.

### 2.1. High-Granular Readout Electrodes

A sampling noble liquid calorimeter is made of absorber plates immersed in a liquid bath. A readout electrode also providing high voltage is inserted in the middle of two consecutive absorbers' plates. Ionization electrons created in the liquid of the two gaps by the showering of the particles therefore drift in the electric field towards the high-voltage pads of the readout electrode. The electrical signal created capacitively during the drift on the readout plane for a given calorimeter cell is routed to the back or front of the electrode, to be amplified and shaped by dedicated readout electronics. The ATLAS LAr calorimeter uses a simple copper-kapton electrode, where signals are routed on the same plane as where they are read out capacitively. To achieve a ten-fold increase in the granularity of the calorimeter compared to ATLAS (i.e., reaching a few million cells) while minimising gaps in the angular coverage, the signals should be routed in a separate layer from the capacitive readout.

This calorimeter concept for FCC-ee therefore uses multilayer printed circuit boards (PCB) to achieve this goal. A schematic side view of the readout is given in Figure 1. The signals are routed to the back of the electrode by traces located in the middle of the PCB, which are connected to the readout plane through vias. As the signal from a given cell travels below the readout planes of all cells located behind it, cross-talk should be prevented by the addition of a shielding (ground) layer between the traces and the readout. The outermost layers provide the high voltage and are capacitively coupled to the readout layer.

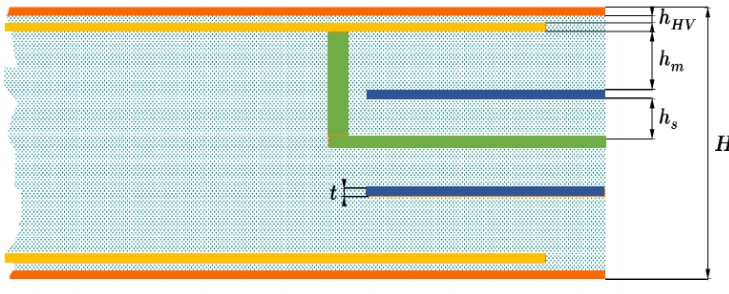

Side view

**Figure 1.** Side view of a readout electrode. The innermost layer contains the readout traces. Above and below it are shields to reduce cross-talk with other cells. The next layer is the readout layer, connected to the trace through vias. The outermost layer provides the high voltage.

The use of PCBs as readout electrodes drives the design of the concept of the electromagnetic calorimeter barrel shown in Figure 2. In order to have a full coverage in $\phi$ with good uniformity and no cracks, the absorber and readout planes are inclined by an angle of 50 degrees around the barrel. The overall dimensions (inner radius of 210 cm and outer radius of 270 cm including the cryostat) are chosen so that the detector can fit in a modified version of the IDEA detector concept [2]. This baseline design uses 2 mm lead plates as absorber and LAr as active material, with gaps ranging from about 2.5 mm at the inner radius to about 4.5 mm at the outer radius. The use of PCBs gives complete freedom in the drawing of the calorimeter cells and us allows therefore to create projective cells along $\theta$ and $\phi$ and to optimise the granularity for physics performance (similarly to the very fine $\eta$ "strips" of the first layer of the ATLAS LAr calorimeter). This baseline design

features 12 longitudinal layers and cells of about $2 \times 2 \, \text{cm}^2$, with a finer segmentation in the second layer. The LAr gap widening effect between the inner and outer radius, which could significantly increase the constant term of the resolution, is compensated by the presence of the 12 longitudinal layers [6]. The preliminary design of the endcap calorimeters use readout and absorber planes perpendicular to the beam axis.

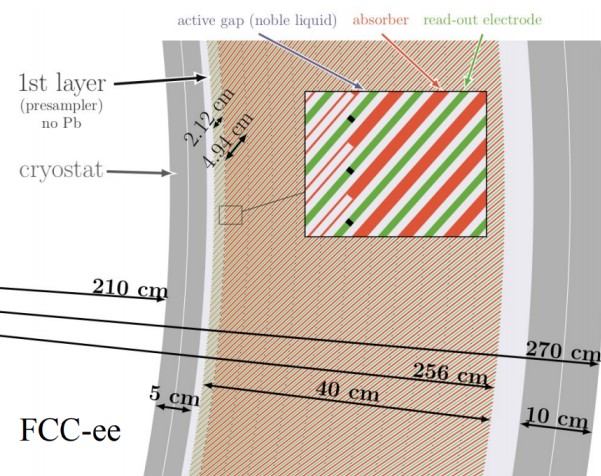

**Figure 2.** Sketch of the transverse view of the design of the electromagnetic calorimeter barrel. Only a slice of the $\phi$ coverage is shown.

Studies are ongoing on the design of the readout electrodes, with simulations and first prototypes, to optimise the performance. Indeed, the heights of the insulation layers between the copper layers and the widths of the traces and the shields, have a strong impact on the cell impedance, on the cell capacitance (and therefore the noise), and on the cross-talk. Simulations are performed both with the Sigrity tool of Cadence and with ANSYS HFSS. The first results show promising performance, with cross-talk limited to below the two per-mille level when using the integration time of a few hundred ns. The cell capacitances are dominated by the capacitance between the readout plane and the shields, indicating there is a trade-off between noise and cross-talk. The first validation of the accuracy of the simulations has been performed by measuring simple test structures (single transmission lines in a PCB). Reasonable agreement between the measurements and the simulation is achieved over the frequency range of interest: about 5 to 10% on the impedance between 1 MHz and 500 MHz. A second series of prototypes is being fabricated to study the impact of the size of the shields and of the depths of the layers and to conduct performance measurements of cells in a full-scale design.

### 2.2. High-Density Feedthroughs

A calorimeter with a few million cells implies that a few million signals have to be extracted from the cryostat. High-density feedthroughs therefore have to be developed, aiming for a density five times greater than that of ATLAS feedthroughs. The new concept being studied avoids the use of connectors. Instead, kapton cables can be slid into 3D-printed epoxy resin structures with slits. They are then glued in place, and the structure is fixed on the stainless steel flange with the use of a bolted compression plate. Leak and pressure tests have been performed at 300 K and 77 K with several designs and glues. Suitable materials showing negligible leaks after thermal cycles have been identified. The design of a complete flange is also making progress. Simulations of stress and deformations at 300 K and 77 K have been performed and allowed us to narrow down adequate solutions.

### 2.3. Cryostat

The cryostat is a crucial element of a noble liquid calorimeter, as it must sustain mechanically the body of the calorimeter, while its front wall should be as thin as possible

to allow the measurement of low-energy particles in the calorimeter. Ongoing directions for the R&D involve new materials, in particular carbon fiber, and sandwiches of materials, benefiting from recent progress in the aerospace industry. These developments are expected to be used in all future detectors, as such cryostats will be built around the future solenoids. In particular, a wall made of a honeycomb aluminium structure sandwiched between two layers of carbon fiber can be robust enough while keeping a material budget of 0.04 radiation lengths ($X_0$) [7]. Ongoing developments at CERN aim to address issues specific to our field. One is that of sealing methods, as the cryostats have to be closed after inserting large structures such as solenoid magnets or a calorimeter. Leak and pressure tests, as well as microcrack resistance tests, are being performed on test structures using Belleville washers for bolting carbon fiber walls together. A second ongoing development addresses the interface between metal and carbon fiber, which is crucial for mounting feedthroughs on top of a carbon fiber cryostat.

## 2.4. Readout Electronics

The energy resolution of a calorimeter is the sum of a constant term, a sampling term, and a noise term. At FCC-ee, where there is no pile-up noise as in the LHC, the latter one is dominated by the electronics noise. The goal is therefore to minimise the noise of the readout chain to achieve a good energy resolution even on low-energy photons (around 200 MeV) and to measure the energy of an MIP with a good signal-to-noise ratio. The dominant noise term goes as $C\sqrt{4kT/(g_m\tau_p)}$, where $C$ is the capacitance that depends on the cell capacitance and on the transmission line, $g_m$ is a characteristic of the transistors in a given technology, and $\tau_p$ is the peaking time of the signal after shaping. Compared to the ATLAS LAr calorimeter case, this design features smaller cells (and therefore smaller $C$), and shaping times can be much longer because of the low number of interactions per bunch crossing (typically 200 ns instead of 50 ns).

Analytical simulations show that a readout chain similar to the one used in ATLAS, where all signals are routed outside of the calorimeter before being amplified and processed by warm electronics, could provide adequate performance, with a noise of a few MeV per cell, and a signal-to-noise ratio of about 3 for the energy of a MIP traversing a cell. An attractive alternative is to use cold electronics located in the cryostat at the back of the calorimeter for at least the amplification and shaping of the signals. This would reduce $C$, as transmission lines are much shorter, $T$ is by definition lower, and the $g_m$ of the transistors is larger. Simulations show that the noise could be reduced by a factor of 5 or more, making it negligible for all measurements and achieving a very high signal-to-noise ratio for MIP deposits. In addition, processing signals in the cold would simplify the requirements on the cryostats, at least on cross-talk, and possibly on signal density if multiplexing can be used. With very low radiation levels expected at FCC-ee, the ageing effects and failures of components in the cold are not expected to be an issue; however, power consumption and heat dissipation should be investigated to make this option possible.

## 2.5. Towards a First Prototype

As developments are ongoing on many technical aspects simultaneously, a natural goal is to build a small prototype of around $40 \times 40$ cm within a few years to prove in testbeam the feasibility of the whole concept. Mechanics studies are expected to start in autumn 2022, with the design of absorbers and spacers. The readout electronics of a first prototype will have to reuse existing components and avoid designing a whole new chain. Fortunately, the requirements of this concept are not far from those of other projects. The readout electronics developed for the DUNE experiment [8] can be used in the cold and could therefore be used for a first prototype, with the only caveat that their dynamic range cannot cover the charge expected from the highest-energy deposits. Conversely, the SKIROC ASIC developed for CALICE Si-W calorimeters [9] matches the physics requirements of this design but was not designed to work in an LAr cryogenic environment. It could, however, be used quite readily for a warm readout electronics option. For a first prototype, it is expected to reuse

an existing aluminium cryostat or possibly a first prototype of a carbon fiber cryostat built at CERN, which may be ready on a similar timescale.

## 3. Expected Performance

In parallel with the R&D studies on the important items for the future realization of this concept, the expected performance of the design was evaluated in simulation. The baseline geometry of the barrel calorimeter was implemented in the FCC software suite using DD4HEP [10] and features 12 layers amounting to about 22 $X^0$. The layers are composed of 2 mm lead absorber plates, 1.2 mm for the readout PCBs, and a double LAr gap of $2 \times 1.2$ mm at the inner radius. The plates were inclined by about 50° around the barrel. The resulting typical calorimeter cell size was about $2 \times 2 \times 3$ cm$^3$. Simple fixed-size clusters' reconstruction and cluster-level corrections enabled the first performance studies.

The main goal of the performance studies was to optimise the design and guide important design decisions: the choice of the absorber (lead vs. tungsten), choice of active material (LAr vs. liquid krypton), and optimisation of the granularity of the cells. The electromagnetic energy resolution was evaluated on single unconverted photon simulations. The relative resolution of the baseline design was $8\%/\sqrt{E} \oplus 0.7\%$, as shown on Figure 3. Work is ongoing to evaluate the expected resolution of alternative choices for absorbers and active material. The use of PCBs as readout electrodes allows large flexibility in the size of the calorimeter cells, possibly using different sizes per layer. The first studies on $\pi^0$ identification efficiency and on classification efficiency of $\tau$ decay modes show promising performance and point towards using small cells in the first calorimeter layers and towards the use of liquid krypton to take advantage of its smaller Moliere radius compared to LAr.

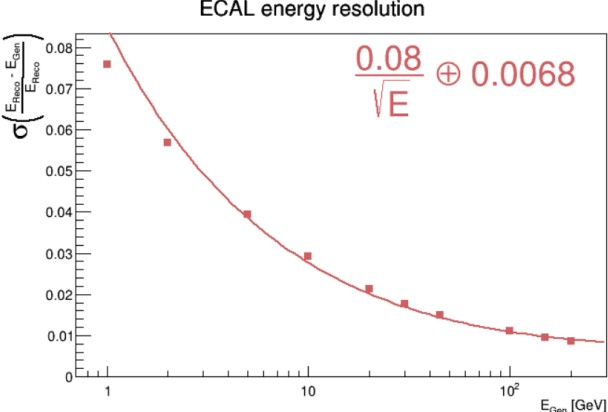

**Figure 3.** Unconverted photon energy resolution in the baseline design using LAr as active material and lead absorbers. No noise is included in the simulation, but it is expected to be negligible compared to the sampling term even for low-energy photons. The constant term is in particular due to an incomplete containment of the showers in the fixed size clusters, and it is expected to be reduced with future improvements in the reconstruction.

The next main milestone of the simulation studies is the study of jet physics, as a jet energy resolution of about 4% at 50 GeV is one of the main requirements for Higgs physics at FCC-ee. Jet reconstruction at future $e^+e^-$ colliders will, however, rely on particle flow algorithms to fully exploit the performance of all detectors. To that end, the Pandora PFA reconstruction [11] is being integrated in the FCC software chain. Once a hadronic calorimeter is added to the detector concept (possibly using one of the CALICE designs), the jet performance of this electromagnetic calorimeter design can be studied and end-to-end detector optimisation can be performed, and in particular the optimal size of cells can be further studied.

## 4. Conclusions and Perspectives

Noble liquids have proved to be an excellent technology for electromagnetic calorimeters, featuring very good energy resolution, granularity, linearity, uniformity, and stability over time. They are therefore appealing candidates for detectors at FCC-ee, with several ongoing R&D efforts recently begun to show the feasibility of future detectors. Evidence is being gained that a high-granularity noble liquid calorimeter could be a feasible and versatile solution, fulfilling the stringent FCC-ee requirements. Good progress is being made on the design of high-granularity readout electrodes and on high-density feedthroughs. The concept takes advantage of the R&D on thin cryostats that will benefit all future experiments. The first simulations show that adequate performance on the electromagnetic objects can be achieved.

Time and resources permitting, other fields of R&D could be explored around this concept to possibly improve its performance. The doping of noble liquid, which increases the signal yield by enhancing the drift velocity, could be studied if the noise needs to be further reduced. The cell-level timing capabilities of the design could be also explored. The readout electronics can easily be optimised, but the performance will be limited by the Landau fluctuations in the sampling, and the usefulness of timing in the reconstruction will depend on the overall detector design and the presence (or the lack thereof) of particle identification detectors using $dE/dx$, timing, or Cerenkov light. Finally, the collection and use of noble liquid scintillation light could be studied. This fast signal is used in dark matter noble liquid detectors, but its use in a calorimeter at a collider would be a novelty and provide a complementary time and energy measurement. There are, however, significant challenges to overcome in order to collect and measure the scintillation light.

**Funding:** This research was funded through the AIDAInnova programme by the European Union's Horizon 2020 Research and Innovation programme under Grant Agreement No. 101004761.

**Data Availability Statement:** Not applicable.

**Conflicts of Interest:** The author declares no conflict of interest.

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
