# Peer review of "Noble Liquid Calorimetry for FCC-ee"

_instruments, doi:10.3390/instruments6040055_

Round 1
Reviewer 1 Report
Good general review of ongoing R&D of electromagnetic calorimetry for the FCC-ee.
Ready for publication after minor revision.

Author Response
Dear reviewer,
Many thanks for your comments, which definitely helped to improve the quality of the manuscript.
They have all been readily implemented.
Best regards,
Nicolas Morange
Reviewer 2 Report
The article describes the path for a noble liquid calorimeter at the FCCee. It identifies the Liquid Argon (LAr) technology as suitable for matching the physics program envisaged while it underlines the substantial R&D needed for an essentially new LAr calorimeter design. It explains how the need for high granularity at future electron-positron colliders can be satisfied with a new readout scheme based on printed circuit boards immersed in the LAr bath. This solution represents a significant challenge with respect to the ATLAS LAr Barrel. The author details the R&D challenges for a first demonstrator including high-density feedthroughs, ultra-light cryostats, and readout electronics. Results on electromagnetic energy resolution based on preliminary simulations are described as well. Overall the article is well written and clear. I suggest accepting it after minor revision.
Minor revision:
Line 5: reqdout -> readout
Lines 23 and 39: References [2] and [3] appear in the text after references [4] and [5]. Please correct the reference number in the references list.
Line 77: The author identifies the longitudinal change in the sampling fraction as a potentially dangerous design and suggests that the presence of 12 longitudinal layers can compensate for its effect. I believe the sentence does not stress enough the implications of such a design. Having 12 longitudinal readout layers with different sampling fractions requires a correction for the different calorimeter responses using 12 calibration constants. However, the electromagnetic shower composition change within a shower with the shower tails typically exhibiting a lower response with respect to the shower core. On top of that the shower's longitudinal shape changes with the primary particle energy. What I would expect is that the 12 calibration constants will depend on the shower energy and, in the case of photons, on the showering starting point. I wonder if this problem has been investigated with simulations. If not, I would suggest a more conservative approach while describing the problem.
Line 116: If possible a reference for the 0.04 X0 material budget would be appreciated.
Line 155: will have a to reuse -> will have to reuse
Line 171: 50^0 typo in deg symbol
Line 177: single-photon simulation -> single unconverted photon simulation
Line 214: The author writes that the collection of scintillation light from LAr "provide some kind of dual-readout for hadronic energy measurement". I find this sentence confusing. Dual-readout measurements for hadronic energy reconstruction involve the measurement of a first signal sensitive to the ionization component in hadronic showers and a second signal sensitive to the electromagnetic component of the hadronic shower as, for instance, the Cherenkov light. In LAr the scintillation signal and the free-charge induced signal are both sensitive to the ionization component of a hadronic shower. So the dual-readout method for hadronic measurements is not applicable in this respect. Therefore I completely miss the point on the benefit of scintillation signal collection for hadronic energy measurement. I suggest rephrasing this sentence.
Author Response
Dear reviewer,
Many thanks for your comments, which definitely helped to improve the quality of my manuscript. Please find some answers below.
Best,
Nicolas Morange
- Regarding the 12 layers. It has been shown on the FCC-hh calo concept that with >=8 layers the gap widening has a negligible effect on the resolution, with calibration constants shown not to depend on the incident particle energy. A reference to [arXiv:1912.09962] has been inserted in the text.
- Ref for 0.04 X0 claim: all that is public are presentations shown at FCC workshops. There are no papers / arxiv available to my knowledge. I therefore cited one of such presentations.
- Scintillation light: your comment is fully correct. The sentence has been changed and toned down to refer to a 'complementary time and energy measurement'.
- All other comments are implemented.